# A myristoyl switch at the plasma membrane triggers cleavage and oligomerization of Mason-Pfizer monkey virus matrix protein

**Markéta Častorálová[1†], Jakub Sýs[1,2†], Jan Prchal[1†], Anna Pavlů[1], Lucie Prokopová[1], Zina Briki[1], Martin Hubálek[2], Tomas Ruml[1]\***

[1]Department of Biochemistry and Microbiology, University of Chemistry and Technology, Prague, Czech Republic; [2]Institute of Organic Chemistry and Biochemistry of Czech Academy of Science, Prague, Czech Republic

**Abstract** For most retroviruses, including HIV, association with the plasma membrane (PM) promotes the assembly of immature particles, which occurs simultaneously with budding and maturation. In these viruses, maturation is initiated by oligomerization of polyprotein precursors. In contrast, several retroviruses, such as Mason-Pfizer monkey virus (M-PMV), assemble in the cytoplasm into immature particles that are transported across the PM. Therefore, protease activation and specific cleavage must not occur until the pre-assembled particle interacts with the PM. This interaction is triggered by a bipartite signal consisting of a cluster of basic residues in the matrix (MA) domain of Gag polyprotein and a myristoyl moiety N-terminally attached to MA. Here, we provide evidence that myristoyl exposure from the MA core and its insertion into the PM occurs in M-PMV. By a combination of experimental methods, we show that this results in a structural change at the C-terminus of MA allowing efficient cleavage of MA from the downstream region of Gag. This suggests that, in addition to the known effect of the myristoyl switch of HIV-1 MA on the multimerization state of Gag and particle assembly, the myristoyl switch may have a regulatory role in initiating sequential cleavage of M-PMV Gag in immature particles.

**\*For correspondence:**
tomas.ruml@vscht.cz

[†]These authors contributed equally to this work

**Competing interest:** The authors declare that no competing interests exist.

## Editor's evaluation

This valuable study advances our understanding of how the viral protease in a D-type retrovirus is activated and in particular how the exposure of the myristoyl group is required for processing of the Gag matrix precursor. The supporting evidence is convincing and suggests that M-PMV proteolytic maturation is triggered by a myristyl switch that occurs when the matrix (MA) domain of Gag interacts with the PM. This manuscript is of interest to retrovirologists and structural biologists.

## Introduction

Interaction with the plasma membrane (PM), budding, and maturation of viral particles are the final steps in retroviral replication cycles. In most retroviruses, the Gag–PM interaction is mediated by a bipartite motif consisting of a highly basic region and the N-terminal myristoyl (myr) moiety of MA. Despite similarities in the main mechanisms of viral capsid assembly and membrane targeting, there are remarkable morphogenetic differences between retroviruses. One significant difference lies in the location of the assembly sites; these are located at the PM for C-type retroviruses such as HIV and in the cytoplasm for D-type retroviruses, such as Mason-Pfizer monkey virus (M-PMV). The

pre-assembled intracytoplasmic viral particles (ICAPs) of D-type retroviruses are then transported to the PM for budding (*Rhee and Hunter, 1990*).

Retroviral MA proteins are the key players in targeting Gag to the PM. HIV-1 MA is myristoylated and exists in two conformational states dictated by its location. In the cytoplasm, the myristoyl (myr) moiety is sequestered inside the hydrophobic pocket of MA. Upon Gag interaction with the PM, myr is expelled from the MA molecule and inserted into the lipid bilayer by the so-called myr switch (*Tang et al., 2004*). In HIV-1, the myr exposure is triggered by the interaction of the MA domain of Gag with phosphatidylinositol-4,5-bisphosphate [PI(4,5)P$_2$], which is present exclusively in the PM (*Ono et al., 2004*; *Saad et al., 2006*; *Tang et al., 2004*). The equilibrium between exposed and sequestered myr states in HIV-1 MA is concentration and oligomerization dependent, in contrast to myristoylated HIV-2 MA, which is exclusively monomeric in vitro. This can be explained by the tighter sequestration of the myr chain within the protein core (*Saad et al., 2008*). In D-type retrovirus M-PMV, MA myristoylation is also essential for the targeting of immature particles to the PM (*Rhee and Hunter, 1991*), but with a significantly lower affinity for water-soluble PI(4,5)P$_2$ compared to HIV-1 MA. In addition, the interaction with this PM component fails to induce the myristoyl switch in purified M-PMV (myr+)MA in vitro (*Kroupa et al., 2016*; *Prchal et al., 2012*).

After the interaction of Gag or pre-assembled viral particles with the PM and budding, maturation occurs. The proteolytic maturation involves a delicately regulated sequential processing of polyprotein precursors (*Konvalinka et al., 2015*). The importance of sequential cleavage of HIV-1 Gag to liberate the mature structural proteins matrix (MA), capsid (CA), nucleocapsid (NC), and p6 and the spacer peptides SP1 and SP2 has been recognized for decades (*Erickson-Viitanen et al., 1989*; *Pettit et al., 1994*; *Tritch et al., 1991*). However, our understanding of how this coordinated event is regulated remains incomplete.

Recently, the challenge of analyzing a population of viruses in heterogeneous life cycle phases was overcome with an approach using a photo-destructible HIV protease inhibitor (*Schimer et al., 2015*). This allowed synchronization of the immature virus particles and triggering of polyprotein processing with light. However, it also uncoupled maturation from assembly, preventing study of the interplay between these processes and the impact of Gag–membrane interactions on proteolytic cleavage. Recently, experiments using nanoscale flow cytometry and instant structured illumination microscopy demonstrated that activation of HIV-1 protease occurs during viral assembly prior to release of the virus (*Tabler et al., 2022*). However, fluorescence lifetime imaging microscopy and single-virus tracking revealed that there is a delay between HIV-1 particle assembly and maturation (*Qian et al., 2022*). The importance of precise regulation of maturation was demonstrated by overexpression of HIV-1 protease which, in addition to its cytotoxic effects, prevented HIV-1 particle formation and budding (*Karacostas et al., 1993*). Similar effect was also shown for Rous sarcoma virus (RSV) where premature proteolytic processing also lead to budding defects (*Xiang et al., 1997*).

In vitro data suggested that initial cleavage of HIV-1 Gag occurs at the SP1/NC site, followed by cleavage at the SP2/p6 and MA/CA sites, and finally at the NC/SP2 and CA/SP1 sites (*Pettit et al., 1994*; *Wiegers et al., 1998*; *Wondrak et al., 1993*). Researchers observed a similar pattern of ordered processing in infected cells by using mutants in which individual cleavage sites were disrupted by point mutations (*Wiegers et al., 1998*). These data led to the conclusion that the cleavage events at individual sites occur with different kinetics. The initial fast cleavage at the C terminus of SP1, releasing NC, allows condensation of the ribonucleoprotein core. Next, CA is liberated from the MA, and finally, the release of SP1 from CA initiates capsid condensation and core formation. Interestingly, even a small proportion of uncleaved Gag has a dominant negative effect on HIV-1 infectivity, as shown by *Müller et al., 2009*. Blocking cleavage of MA/CA site had the most pronounced impact, exhibiting a transdominant negative effect (*Lee et al., 2009*). Both MA–CA and MA–CA–SP1 maturation products accumulated in cells when the MA/CA cleavage site was blocked by point mutations (*de Marco et al., 2010*; *Mattei et al., 2018*; *Müller et al., 2009*). Similarly, in murine leukemia virus, partially cleaved Gag interferes with infectivity when present at equimolar concentrations with the mature proteins (*Rulli et al., 2006*).

In M-PMV, the protease domain of Gag-Pro and Gag-Pro-Pol polyproteins can theoretically dimerize in ICAPs, however, processing is not initiated until the particles reach the PM (*Parker and Hunter, 2001*). Thus, the maturation of D-type retroviruses must be tightly regulated and the mere Pro dimerization is probably not sufficient for effective ICAP maturation. In vitro analysis of the proteolytic

processing of intracytoplasmic M-PMV particles also suggested a sequential manner of Gag maturation. However, the first protein domain released by M-PMV protease from Gag was MA, with a 10-fold greater affinity for the MA/PP cleavage site than other Gag cleavage sites (*Pichová et al., 1998*). *Parker and Hunter, 2001* observed similar cleavage kinetics for M-PMV MA/PP and PP/p12 cleavage sites in COS-1 cells. This is in contrast with data on HIV-1, for which initial cleavage occurs at SP1/NC (*Wiegers et al., 1998*), and RSV, for which in vivo studies showed that the MA–p2 junction is the most slowly cleaved site (*Xiang et al., 1997*). This was supported by in vitro cleavage of peptides mimicking the cleavage sites in RSV Gag, in which both the MA–p2 and p2–p10 junctions were cleaved very inefficiently compared to the other sites (*Cameron et al., 1992*).

Interestingly, the HIV-1 (myr+)MA V7R mutant, in which myr is sequestered in the MA core, was shown to be defective in Gag processing in HeLa and 293T cells (*Hikichi et al., 2019*; *Ono and Freed, 1999*). This was confirmed in CD4(+) T cells, which are the natural target cells of HIV-1 (*Lee et al., 1998*). In M-PMV, the possible role of MA myristoylation in maturation was also suggested (*Prchal et al., 2011*). Even though M-PMV MAPP (matrix-phosphoprotein) (MA C-terminally extended with part of the downstream phosphoprotein [PP] domain of M-PMV Gag) can be cleaved efficiently by M-PMV protease in vitro, the N-terminally myristoylated MAPP [(myr+)MAPP] is cleaved very poorly (*Prchal et al., 2011*). Furthermore, the NMR (nuclear magnetic resonance) solution structure of myristoylated MAPP showed a short alpha helix spanning the position of the MA/PP cleavage site that is closely associated with the myristoyl moiety (*Prchal et al., 2012*).

Data presented in this study indicate that the interaction of M-PMV Gag with the PM can significantly enhance both trimerization and cleavage of MA from the rest of Gag, suggesting it may serve as an additional mechanism for maturation control. More broadly, we demonstrate that the interaction of the N-terminal myristoyl with the PM can be projected to the C-terminal region of MA to increase the availability of the domain for downstream processes such as proteolytic cleavage.

## Results

### Interaction of M-PMV (myr+)MAPP with liposomes triggers MA myristoyl switch and subsequent proteolytic processing at the MA/PP junction

M-PMV maturation occurs upon the interaction of M-PMV ICAPs with the PM during and after budding. However, as we have already shown, the myristoylated M-PMV MA extended with 18 amino acid residues from the downstream PP domain of Gag and a His-tag anchor [ (myr+)MAPP] is cleaved very poorly by M-PMV protease (*Prchal et al., 2011*). The mechanism that could trigger proteolytic processing of membrane-bound myristoylated Gag is the exposure of myristoyl from the hydrophobic pocket of MA, the myristoyl switch. Therefore, we sought to confirm that interaction with liposomes enables cleavage at the MA/PP junction in (myr+)MAPP. We compared the efficiency of cleavage of both M-PMV (myr−)MAPP and (myr+)MAPP bound to liposomes mimicking the phospholipid composition of the PM inner leaflet (*Doktorova et al., 2017*). As previously observed (*Prchal et al., 2011*), in the absence of liposomes, (myr−)MAPP was quickly cleaved by M-PMV protease (*Figure 1A*), but cleavage of (myr+)MAPP was significantly delayed (*Figure 1B*). Our new data reveal that binding of (myr+)MAPP to liposomes significantly enhances the cleavage kinetics (*Figure 1B*). Therefore, we propose that the interaction of (myr+)MAPP with liposomes triggers the myristoyl switch that subsequently enables efficient cleavage at the MA/PP junction of M-PMV (myr+)Gag.

### The affinity of myristoyl for the hydrophobic pocket of MA modulates MA cleavage from the rest of Gag

To study the effect of myristoyl switch on the accessibility of (myr+)MAPP C-terminus to the M-PMV protease, we prepared mutants that either facilitate myristoyl release (myrOUT) or prevent the switch (myrIN) by reducing or enhancing the hydrophobicity of the protein core, respectively. To design the 'myrOUT' mutant, we identified large hydrophobic amino acid residue I51 which is in direct contact with the myristoyl in M-PMV (myr+)MAPP (9) and substituted it with alanine (I51A) (*Figure 1—figure supplement 1*). For the 'myrIN' mutant, we introduced the A79V mutation previously suggested to affect myristate accessibility in M-PMV (myr+)MA (36) (*Figure 1—figure supplement 1*). None of these mutations lie near the cleavage site of the MA/PP junction, to exclude the possibility of its direct

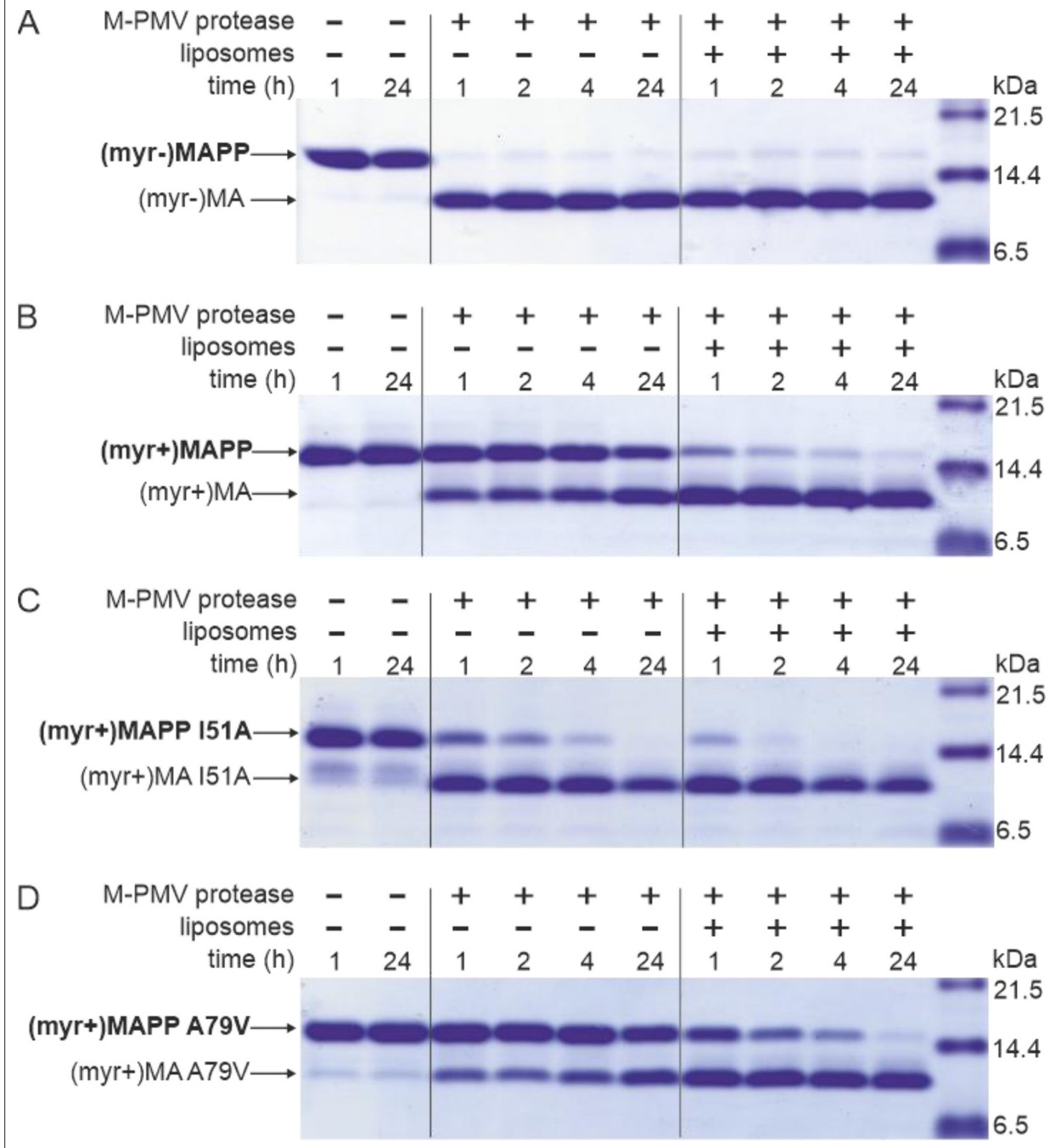

**Figure 1.** MAPP cleavage by Mason-Pfizer monkey virus (M-PMV) protease. (**A**) (myr−)MAPP, (**B**) (myr+)MAPP, (**C**) 'myrOUT' mutant (myr+)MAPP I51A, and (**D**) 'myrIN' mutant (myr+)MAPP A79V were cleaved by M-PMV protease in the absence or presence of liposomes mimicking the plasma membrane (PM) inner leaflet. All the experiments were performed in three biological replicates with the same results. Source data – *Figure 1—source data 1*.

The online version of this article includes the following source data and figure supplement(s) for figure 1:

**Figure supplement 1.** Structure of wild-type (WT) (myr+)MAPP indicating positions of the mutated residues.

**Figure supplement 2.** Comparison of (myr+)MAPP wild-type (WT) and mutants HN-HSQC (1H–15N-heteronuclear single quantum coherence) spectra.

**Source data 1.** Source data for *Figure 1* containing sodium dodecyl sulfate–polyacrylamide gel electrophoresis (SDS–PAGE) gels.

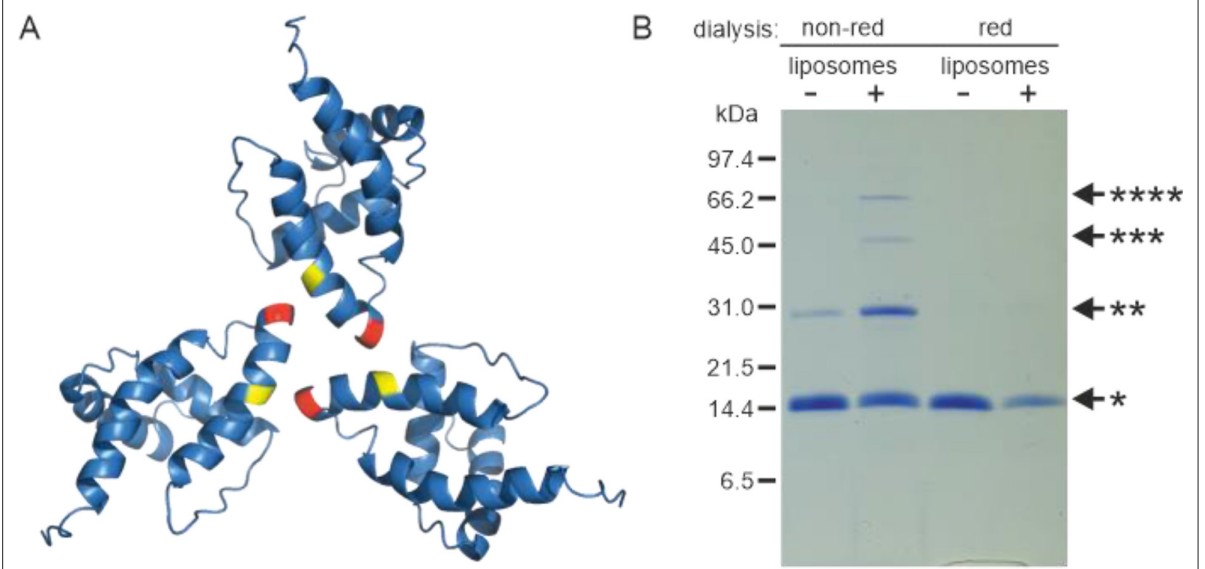

**Figure 2.** Oligomerization of (myr−)MAPP. (**A**) Previously published structure of wild-type (WT) (myr−)MA trimer (*Vlach et al., 2009*) with residues 62 (yellow) and 69 (red) highlighted. (**B**) Sodium dodecyl sulfate–polyacrylamide gel electrophoresis (SDS–PAGE) gel showing oligomers of T69C (myr+) MAPP formed upon the interaction of T69C (myr+)MAPP with liposomes, stabilized by disulfide bridges under non-reducing conditions. non-red – non-reducing conditions; red – reducing conditions; *monomer, **dimer, ***trimer, and ****tetramer. Source data – *Figure 2—source data 1*.

The online version of this article includes the following source data and figure supplement(s) for figure 2:

**Source data 1.** Source data for *Figure 2* containing sodium dodecyl sulfate–polyacrylamide gel electrophoresis (SDS–PAGE) gel.

**Figure supplement 1.** Oligomerization of (myr−)MAPP T69C.

**Figure supplement 1—source data 1.** Source data for *Figure 2—figure supplement 1* containing sodium dodecyl sulfate–polyacrylamide gel electrophoresis (SDS–PAGE) gel.

structural alteration. We also measured HN-HSQC spectra of both (myr+)MAPP mutant proteins and compared them with the WT spectrum. We observed bigger chemical shift changes only for the signals of residues in proximity of the mutation which proves that neither mutation has altered the overall fold of the protein (*Figure 1—figure supplement 2*).

Here, we show that 'myrOUT' (myr+)MAPP mutant was cleaved more efficiently than WT (myr+) MAPP (compare *Figure 1B, C*), but less efficiently than the WT (myr−)MAPP (*Figure 1A*), that fully mimics the 'myrOUT' conformation. The effect of the 'myrIN' substitution in (myr+)MAPP was apparent only upon interaction with liposomes, when preventing the switch should reduce cleavage efficiency compared to the WT (myr+)MAPP. As expected, in the absence of liposomes, both the 'myrIN' mutant (*Figure 1D*) and WT (myr+)MAPP (*Figure 1B*) were cleaved poorly. More importantly, in the presence of liposomes, the cleavage of 'myrIN' mutant was slower than that of WT (myr+)MAPP (compare *Figure 1B and B*). These results show that the myristoyl switch enables cleavage of (myr+)MAPP by M-PMV protease and the sequestered myristoyl delays the cleavage.

## Liposome interaction and myristoyl switch induce (myr+)MAPP oligomerization

Based on the known structure of the M-PMV (myr−)MA trimer (*Vlach et al., 2009*), we designed a mutant allowing covalent cross-linking of monomers to enrich transiently formed MA trimers. We used an approach similar to that of *Tedbury et al., 2016* for HIV-1 MA. They introduced cysteines at positions G62 and S66 of HIV-1 MA, allowing spontaneous disulfide bridge formation between MA molecules. Accordingly, we replaced threonine at position 69 of M-PMV MAPP with a cysteine residue for trimer stabilization. In the M-PMV (myr−)MA trimer, T69 is located directly opposite C62 from the neighboring MA monomer (*Figure 2A*). Thus, its replacement with cysteine allows spontaneous formation of a disulfide bridge under non-reducing conditions and stabilizes the trimeric form of MA.

Under reducing conditions the (myr+)MAPP T69C mutant remained monomeric both in the absence and in the presence of liposomes. Under non-reducing conditions, the (myr+)MAPP T69C mutant remained predominantly monomeric, with a small fraction of dimers. However, in the presence of liposomes under non-reducing conditions, the oligomeric fraction of the protein increased significantly, and dimers, trimers, and tetramers were formed (*Figure 2B*). This suggests that the MA myristoyl switch that occurs after MA interaction with liposomes promotes MA oligomerization. Notably, the WT (myr+)MAPP, which features two surface-exposed cysteines (C42 and C62), does not undergo oligomerization under identical conditions (*Figure 2—figure supplement 1*). This suggests, that random cysteine exposure does not lead to the formation of disulfide bridges in the WT (myr+) MAPP, indicating that the presence of surface-exposed cysteines is insufficient to induce non-specific oligomerization. This is in contrast to the (myr+)MAPP T69C, where oligomerization is supported by the addition of cysteine in position 69 (*Figure 2A* – red) which can form disulfide bridge with cysteine 62 (*Figure 2A* – yellow).

## Myristoyl exposure modulates the accessibility of MA junction with PP

We used hydrogen–deuterium exchange (HDX)-mass spectrometry (MS) to map in detail the observed differences in (myr+)MAPP and (myr−)MAPP behavior. The hydrogen exchange rates of backbone amides which are indicative of their surface exposure and availability were determined at 4°C in the time interval ranging from 2 to 120 s, corresponding to 0.075 to 4.5 s at 37°C. The statistically significant differences ($CI_{98\%}$ = ±0.410 Da) were detected in regions 16–33, 67–85, and 89–110 (*Figure 3—figure supplement 1*, *Figure 3—source data 1*). In *Figure 3B*, the first two regions of (myr−)MAPP showed considerable HDX decrease (*Figure 3B*, blue shaded segments), while the 89–110 regions displayed a significant increase (*Figure 3B*, red shaded segment in deuterium accessibility compared to (myr+)MAPP. The red shaded segment shows that the absence of myristoyl in (myr−)MAPP can destabilize the alpha-helical secondary structure in the K92-L110 region, allowing it to unfold and become more flexible to facilitate proteolytic cleavage.

## Mutation that induces the myristoyl switch increases the dynamics of the (myr+)MAPP structure

To confirm the effect of the myristoyl switch, we monitored differences between deuterium incorporation into the 'myrOUT' mutant I51A and WT (myr+)MAPP (*Figure 3—figure supplement 2*). The I51A amino acid substitution induced a structural transition from (myr+)MAPP to (myr−)MAPP-like (*Figure 3—source data 2*). Specifically, we observed changes in regions 89–110 (*Figure 3C* and C-terminal red shaded segment), well indicated by peptides 90–102 and 101–108, respectively (*Figure 3E*). The peptides 90–102 and 101–108 became more exposed in the I51A (myr+)MAPP mutant compared to WT (myr+)MAPP. The effect of the mutation strongly resembles WT (myr−)MAPP in deuterium exchange rate, as I51A (myr+)MAPP reaches similar values through the whole monitored time course (*Figure 3E*).

## A mutation that prevents the myristoyl switch stabilizes the (myr+) MAPP core and alpha helix in the protease cleavage site

Using HDX-MS, we observed changes in the structural dynamics of the 'myrIN' mutant A79V (myr+) MAPP compared to WT (myr+)MAPP (*Figure 3D* and C-terminal blue shaded segment; *Figure 3—figure supplement 3*; *Figure 3—source data 3*). The A79V mutation considerably stabilized the protein core. The regions 89–110, which harbor the protease cleavage site, initially (after 2 s) exhibited even lower HDX levels in the A79V mutant compared to WT. A gradual increase in deuteration with a plateau reached after 20 s (*Figure 3E*, peptides 90–102) or beyond 120 s (*Figure 3E*, peptides 101–108) for both WT and the A79V (myr+)MAPP mutant reflects slower HDX kinetics caused by secondary structure present in the region comprising the protease cleavage site. In contrast, WT (myr−)MAPP and the I51A (myr+)MAPP mutant reached near-maximal deuteration levels at the initial time point (2 s) and maintained an almost-constant level during the measured time period (*Figure 3E*, peptides 90–102 and 101–108). This suggests the absence of structure in their 89–110 regions.

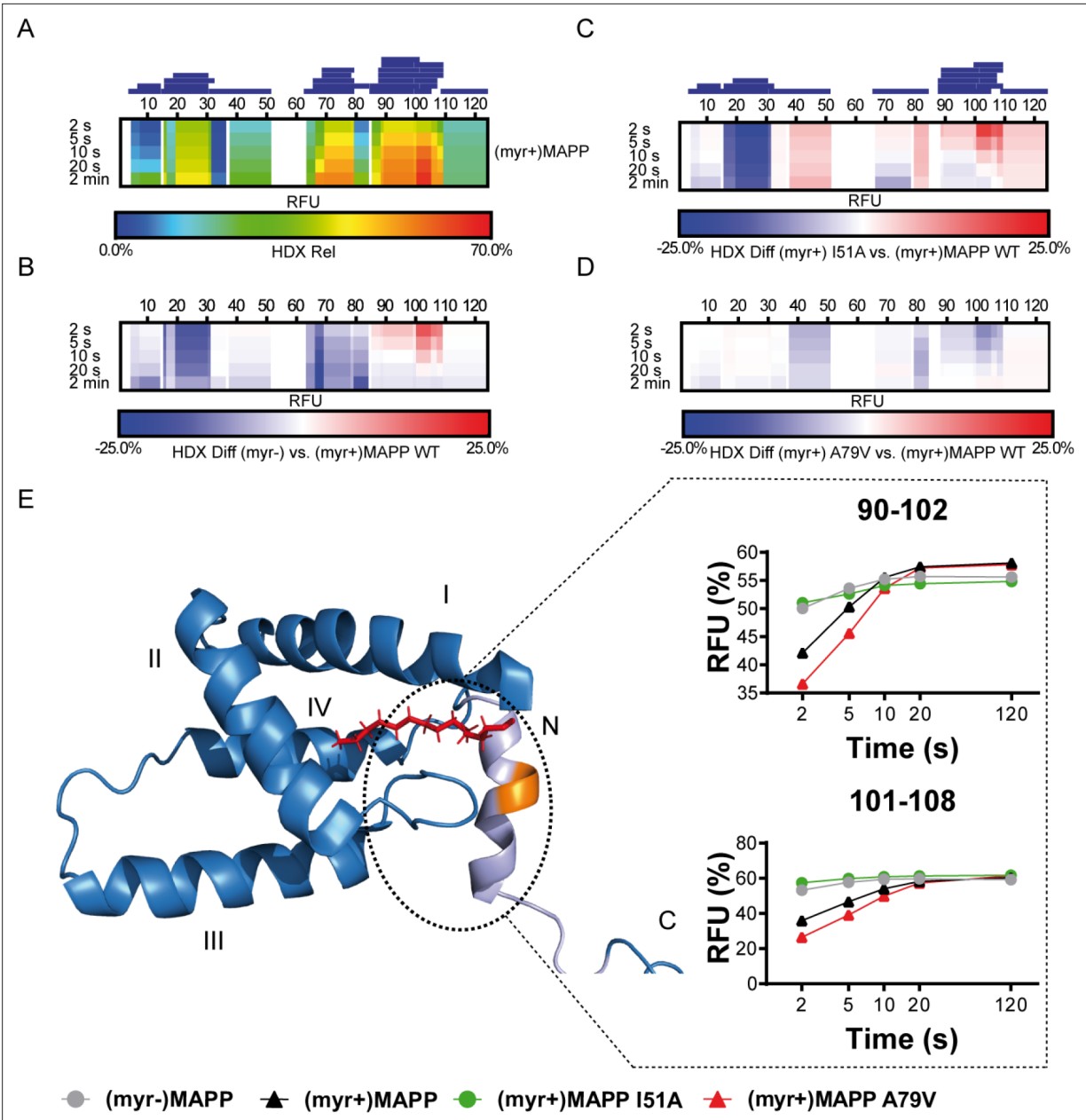

**Figure 3.** Deuterium exchange rates of (myr+)MAPP wild-type (WT) and its comparison to (myr−)MAPP WT and I51A and A79V mutants of (myr+) MAPP. Comparison of hydrogen–deuterium exchange (HDX) rates of backbone amide hydrogens for (myr+)MAPP WT with HDX rates of other protein forms. Panels A–D: Heat maps show the HDX rate of identified peptides indicated as blue rectangles above the amino acid numbering. HDX is shown in rainbow colors (**A**) or in blue–white–red color scale for differential heat maps (**B–D**) as percent relative fractional uptake (RFU, %) at the corresponding time points. Regions without HDX data are shown as white spaces. (**A**) The HDX heat map of (myr+)MAPP WT. (**B**) Differential HDX heat map of (myr−) MAPP WT compared to (myr+)MAPP WT. (**C**) Differential HDX heat map of (myr+)MAPP I51A compared to (myr+)MAPP WT. (**D**) Differential HDX heat map of (myr+)MAPP A79V compared to (myr+)MAPP WT. In differential HDX maps the less accessible amino acids/regions are displayed in blue, while more accessible ones are in red. (**E**) The previously published NMR structure of (myr+)MAPP (***Prchal et al., 2012***, RCSB PDB: 5LMY) with protease cleavage site shown in orange, myristoyl in red, residues 96–110 in light violet, and first four helices of MA are numbered. Graphs show differences in HDX exchange rates in the 90–108 regions around the protease cleavage site represented by peptides 90–102 and 101–108 for different forms of MAPP protein. Related source data – ***Figure 3—source data 1***, ***Figure 3—source data 2***, and ***Figure 3—source data 3***.

The online version of this article includes the following source data and figure supplement(s) for figure 3:

**Figure supplement 1.** Comparison of hydrogen–deuterium exchange (HDX) of (myr+)MAPP and (myr−)MAPP wild types (WTs).

**Figure supplement 2.** Comparison of hydrogen–deuterium exchange (HDX) of (myr+)MAPP and I51A (myr+)MAPP.

*Figure 3 continued on next page*

*Figure 3 continued*

**Figure supplement 3.** Comparison of hydrogen–deuterium exchange (HDX) of (myr+)MAPP and A79V (myr+)MAPP.

**Source data 1.** Comparison of hydrogen–deuterium exchange (HDX) of (myr+)MAPP and (myr−)MAPP – data related to *Figure 3B*.

**Source data 2.** Comparison of hydrogen–deuterium exchange (HDX) of (myr+)MAPP and I51A (myr+)MAPP – data related to *Figure 3C*.

**Source data 3.** Comparison of hydrogen–deuterium exchange (HDX) of (myr+)MAPP and A79V (myr+)MAPP – data related to *Figure 3D*.

## A myristoyl switch modulates the secondary structure of the protease cleavage site between the M-PMV Gag MA and PP domains

Analysis of the previously reported structure (*Prchal et al., 2012*) of (myr+)MAPP identified an alpha helix spanning residues 98–106 that directly interacts with the myristoyl moiety. This region contains the M-PMV protease cleavage site located between residues 100 and 101 at the boundary of MA and PP (*Figure 4A*). Direct comparison with the structure of non-myristoylated MAPP [(myr−)MAPP],

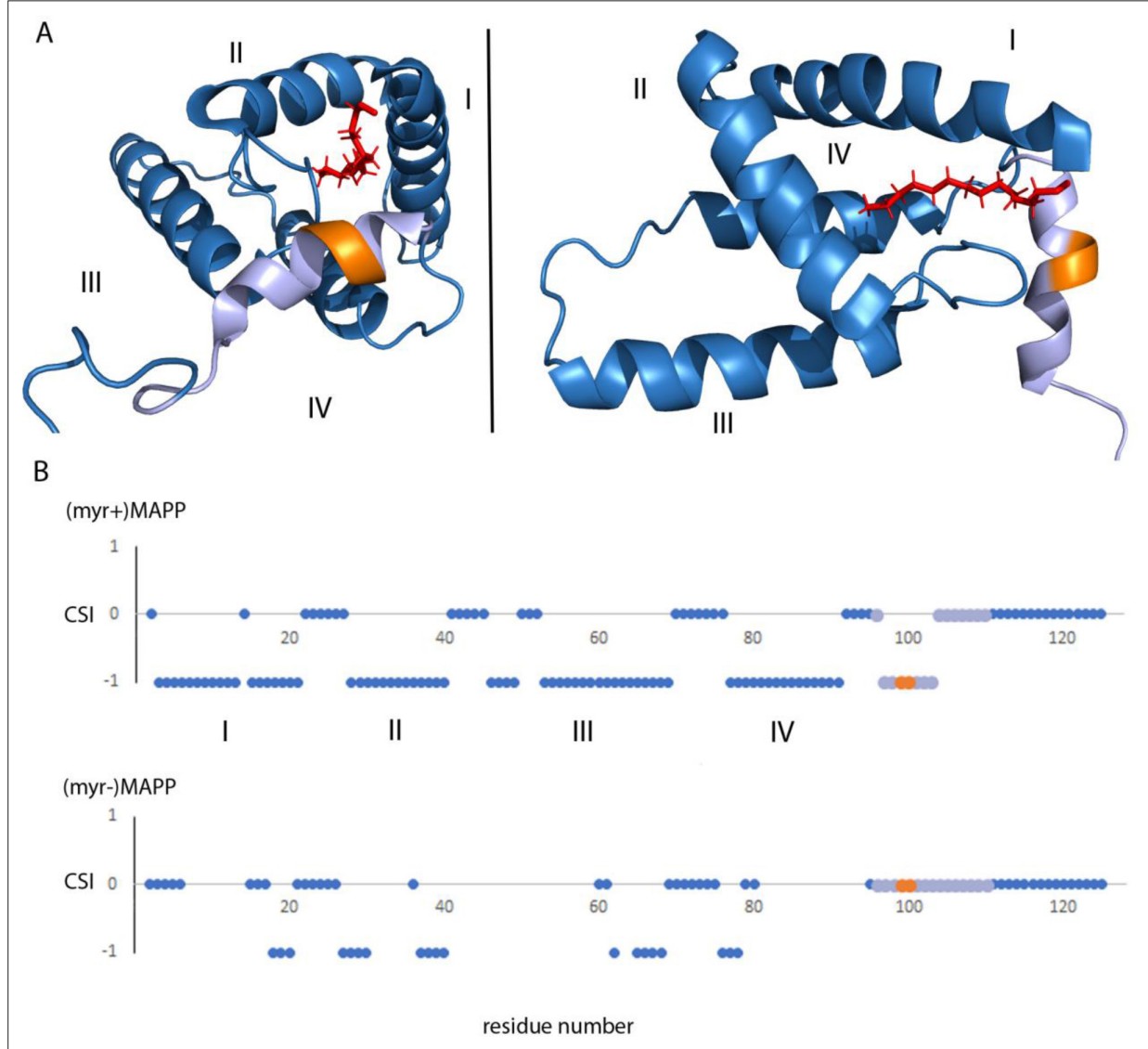

**Figure 4.** Structure of (myr+)MAPP (*Prchal et al., 2012*), position and secondary structure of the cleavage site between MA and PP. (**A**) The protease cleavage site in (myr+)MAPP is shown in orange, myristoyl in red, residues 96–110 in light violet. The first four helices of MA (blue) are numbered, the myristoyl is shown in red. (**B**) Backbone chemical shifts analysis of (myr+)MAPP and (myr−)MAPP proteins using TALOS+ software. CSI (Chemical Shift Index) 0 indicates that residue is in the loop region, CSI −1 indicates that residue is in alpha-helical region. The protease cleavage site is shown in orange, residues 96–110 in light violet. The first four helices of MA (blue) are numbered.

which structurally mimics the protein with exposed myristoyl, was not possible due to the non-specific oligomerization of (myr−)MAPP, that lead to the broadening of the signals from the protein central part. However, we were able to assign the chemical shifts of backbone residues 95–119 in (myr−) MAPP. The chemical shift of the backbone atom of a protein strongly depends on the local structure. We then used TALOS+ software (*Shen et al., 2009*), which compares the observed chemical shifts of backbone atoms with a large database of proteins with known structures to predict the secondary structure of this region (*Supplementary file 1*). This analysis showed that this entire region is unstructured in (myr−)MAPP (*Figure 4B*, *Supplementary file 2*). The same analysis of (myr+)MAPP predicted an alpha-helical region between residues 98 ann− 103. This indicates that the presence of the C-terminal fifth helix in M-PMV MA depends on sequestration of the N-terminal myristoyl in the hydrophobic pocket of the protein.

## Discussion

Here, we examined the hypothesis that myristoyl exposure from the hydrophobic pocket of M-PMV MA triggers a conformational change at the C-terminus, facilitating proteolytic cleavage at the MA/PP junction in the Gag polyprotein. This hypothesis was based on previous results showing that non-myristoylated MA, in contrast to the myristoylated version, is efficiently cleaved from its downstream sequence in M-PMV Gag (*Prchal et al., 2011*). Our basic assumption was that myristoylated M-PMV MA achieves a conformation similar to that of non-myristoylated MA upon the exposure of myristoyl from the MA during the interaction with the PM.

There is a logical basis for proteolytic release of MA from M-PMV Gag to be driven by a myristoyl switch at PM. In D-type retroviruses, pre-assembled ICAPs consisting of N-terminally myristoylated Gag must travel to the PM. Premature intracytoplasmic cleavage of (myr+)MA from Gag or its myristoylation defect would prevent transport of ICAPs to the PM. This was documented by Rhee and Hunter, who showed that mutant non-myristoylated M-PMV particles failed to reach the PM and accumulated in the cytoplasm (*Rhee and Hunter, 1987*).

Unlike in HIV-1, in M-PMV the MA myristoyl switch has not been proved. In HIV-1, the myr exposure is triggered by the interaction of the MA with $PI(4,5)P_2$ (*Ono et al., 2004*; *Saad et al., 2006*; *Tang et al., 2004*) and the equilibrium between exposed and sequestered myr states in HIV-1 MA is also concentration dependent (*Saad et al., 2008*). Recently, it has been shown that upon interaction with the PM, MA induces the insertion of a single $PI(4,5)P_2$ acyl chain into the lipid-binding pocket of MA (*Qu et al., 2021*). However M-PMV MA has significantly lower affinity for water-soluble $PI(4,5)P_2$ and the interaction with this PM component fails to induce the myristoyl switch in purified M-PMV (myr+) MA in vitro (*Kroupa et al., 2016*; *Prchal et al., 2012*).

To prove the myristoyl switch in M-PMV MA and show its possible role in the proteolytic cleavage at MA/PP junction in vitro, we tried to simulate the in vivo situation, by interacting M-PMV MAPP with liposomes mimicking the composition of an inner leaflet of the PM (*Doktorova et al., 2017*). And indeed, we have shown that the interaction of M-PMV (myr+)MAPP with liposomes enabled efficient cleavage of M-PMV (myr+)MA from the downstream PP domain of Gag, probably due to the induced myristoyl switch in (myr+)MAPP. The role of the myristoyl switch itself, rather than the overall interaction of (myr+)MAPP with liposomes, on (myr+)MAPP cleavage was demonstrated using the 'myrOUT' mutant, where the isoleucine residue at position 51, which is in direct contact with myristoyl in the M-PMV (myr+)MAPP (*Prchal et al., 2012*), was substituted by alanine (I51A). This amino acid substitution should facilitate myristoyl release by reducing the hydrophobicity of the protein core. The 'myrOUT' (myr+)MAPP I51A mutant was cleaved more efficiently than WT (myr+)MAPP, but less efficiently than the WT (myr−)MAPP, that fully mimics the 'myrOUT' conformation, confirming the impact of the myristoyl switch on the cleavage. Furthermore, the role of interaction of (myr+)MAPP with liposomes in (myr+)MAPP myristoyl switch was confirmed using the 'myrIN' mutant, which carries the A79V substitution known to impact myristate accessibility in M-PMV (myr+)MA (*Conte et al., 1997*). In the presence of liposomes, the (myr+)MAPP A79V mutant was cleaved more slowly than the WT (myr+)MAPP. This indicates that the interaction of (myr+)MAPP with liposomes activates the myristoyl switch in (myr+)MAPP, allowing for subsequent cleavage.

We used HDX-MS to confirm differences in accessibility of the MA/PP junction in (myr+) MAPP and (myrr−)MAPP. HDX is a valuable method for probing protein conformation and dynamics without perturbation. This technique utilizes MS to monitor the time-dependent incorporation of deuterium

into solvent-accessible backbone amide hydrogens of peptides. Rapid exchange rates of these amide hydrogens suggest flexibility in the surrounding region, whereas slower exchange rates indicate a relatively rigid conformation. Consequently, these exchange rates provide insights into the protein conformation. Although, due to the technical limitations, the data were collected in the absence of membrane, the comparison of the (myr+)MAPP, (myrr−)MAPP, and the mutants reflects the membrane bound and unbound states of the proteins. The only evidence of possible transition between the two conformational states (myrIN and myrOUT) was observed by HDX-MS for I51A mutant. Destabilization of hydrophobic core shifts this mutant phenotypically closer the (myr−)MAPP (*Figure 3E*, peptides 90–102 and 101–108) mimicking the membrane-bound (myr+)MAPP with exposed myristoyl. HDX-MS data showed that the absence of the N-terminal myristoyl in the binding pocket destabilizes the other-wise alpha-helical secondary structure in the K92-L110 region of (myr+)MAPP (*Prchal et al., 2012*), which makes the protease cleavage site more accessible. The effect of the myristoyl switch was further confirmed by analysis of cleavage of (myr+)MAPP protein carrying amino acid substitution I51A. This 'myrOUT' mutant was cleaved more efficiently than WT(myr+)MAPP. HDX-MS analysis documented changes in the region spanning residues 89–110 (*Figure 3B–D*), most prominent in the peptides 90–102 and 101–108 (*Figure 3E*). In contrast, the 'myrIN' A79V (myr+)MAPP mutant with tightly sequestered myristoyl group was cleaved less efficiently in the presence of liposomes compared to WT. The tightly sequestered myristoyl hindered the myristoyl switch in liposome-bound 'myrIN' mutant, slowing the cleavage. Results of HDX-MS analysis also showed a similar, pattern of slowly increasing deuterium exchange rate for 89–110 regions in the 'myrIN' mutant and WT (myr+)MAPP (*Figure 3D*). In both the 'myrIN' mutant and WT (myr+)MAPP, the plateau of deuteration for peptides 90–102 and 101–108 was achieved only after an extended period of 20 s (*Figure 3E*, peptides 90–102), or after 120 s (*Figure 3E*, peptides 101–108). Similarly, the well-cleaved 'myrOUT' mutant was almost deuterium saturated after only 2 s and showed no significant evolution after that, suggesting that the region spanned by peptides 90–102 and 101–108 is highly dynamic and thus accessible for deuterium exchange (*Figure 3E*). The fast HDX kinetics of 'myrOUT' mutant corresponds with its higher suscep-tibility to proteolytic cleavage compared to WT (myr+)MAPP and 'myrIN' mutant. The differences in dynamics reflecting protease cleavage site accessibility could be monitored for up to 5 s with a dramatic difference in the first 2 s. These results support our data suggesting that the sequestered myristoyl participates in the mechanism preventing proteolytic separation of MA from the rest of Gag prior to its interaction with the PM.

Analysis of the secondary structure parameter estimation suggested that the region spanning amino acid residues 98–103 in (myr−)MAPP is unstructured and lacks any periodic secondary struc-ture. In contrast, the region spanning these residues in (myr+)MAPP is ordered into the terminal fifth alpha helix of MA (*Prchal et al., 2012*). Residue 98 directly interacts with the myristoyl moiety, and residues 101 and 102 with the loop connecting helices II and III. The conformation of this loop differs depending on the myristoylation state of MA, as the loop is directly involved in the formation of the myristoyl-holding cavity. Thus, the switch can lead to a large conformational change in this region and destabilize the helix, resulting in exposure of the MA/PP cleavage site. This indicates that the interaction of the MA domain of M-PMV Gag with the PM, triggering the myristoyl switch, may be an important regulatory element in the Gag processing of D-type retroviruses. The MA-embedded myr could prevent premature cleavage of (myr+)MA from the downstream portion of Gag during transport of immature particles to the PM. Some premature cleavage was suggested by previously observed intracytoplasmic protease activation in transfected cell lysates (*Rhee and Hunter, 1990*). However, it remains unclear whether this cleavage occurred in the cytosol or upon the interaction of immature particles with the PM and MA myristoyl switch.

In type C retroviruses, the myristoyl switch promotes Gag anchoring in the PM and thus final steps of particle formation, budding, and maturation (*Hermida-Matsumoto and Resh, 1999*; *Spearman et al., 1997*). A myristoylation defect prevents HIV-1 Gag from PM binding and protease activation in T cells (*Lee et al., 1998*). In contrast to HIV-1 where sequential Gag cleavage was documented (*Pettit et al., 1998*; *Wiegers et al., 1998*; *Wondrak et al., 1993*), no detailed data exist on M-PMV Gag processing kinetics. However, in vitro experiments with immature particles from COS-1 cells showed that the reducing agent induces activation of the M-PMV protease to form MA, PP, and CA (*Parker and Hunter, 2001*). The same authors observed that the M-PMV MA was separated from the rest of Gag in the absence of membranes very inefficiently, aligning with the sequestration of myristoyl in

MA. However, the (myr+)MA/PP junction was cleaved more readily compared to other processing sites.

In addition to aiding Gag processing, myristoyl exposure at the PM modulated the oligomerization of M-PMV MA. Although tertiary structures of MA proteins are very similar across retroviruses, they vary in their primary structures and surface features, including the surface distribution of hydrophilic and hydrophobic residues. In HIV-1, (myr+)MA occurs in a momomer–trimer equilibrium, with the myristoyl sequestered in the monomer and exposed in the trimer (*Tang et al., 2004*). Moreover, MA trimers in HIV-1 particles have been observed in vivo (*Tedbury et al., 2016*). Interestingly, structurally similar HIV-2 MA is monomeric in both its myristoylated and non-myristoylated forms (*Saad et al., 2008*). M-PMV significantly differs from the C-type retroviruses in the MA interaction with membrane components, Gag organization and particle assembly. Some of these properties preclude the use of full-length M-PMV Gag for structural and some functional studies. The obstacles are related to the fact that M-PMV Gag assembles into immature particles in vitro in the absence of membranes (*Klikova et al., 1995*), but HDX and NMR methods are not suitable for working with pre-assembled particles. A possible method for obtaining relevant structural data would be cryo-EM, which was used to obtain the high-resolution lattice structure of MA HIV-1 (*Qu et al., 2021*). However, unlike the distinctive hexagonal lattice formed by HIV-1 MA trimers, the M-PMV MA layer lacks any observable ordered structure (James Stacey, personal communication). This confirms lower oligomerization capacity of M-PMV MA compared to HIV-1 MA. Given the above, the MA–PP fusion protein appears to be the best available model mimicking M-PMV. We also hypothesize that within the overall context of Gag, M-PMV MA retains significantly higher structural flexibility compared to HIV-1 MA. This is because HIV-1 MA is separated from the rigid CA Gag domain layer solely by relatively short flexible linker, in contrast to M-PMV MA which is separated from CA domain by two protein domains, PP and p12.

As mentioned above, unlike HIV-1 (myr+)MA, M-PMV (myr+)MAPP does not oligomerize in vitro (*Prchal et al., 2012*) and (myr−)MA exists in a monomer–dimer–trimer equilibrium (*Srb et al., 2011*; *Vlach et al., 2009*). These findings along with our cleavage experiments, justifies the hypothesis that the myristoyl switch converting M-PMV (myr+)MA into (myr−)MA-like conformation promotes its oligomerization. This suggests that the presence of the myristoyl in the hydrophobic cavity of (myr+) MA affects the hydrophobic interactions occurring at the oligomerization interfaces of the monomers. This was confirmed by using the T69C mutant with stabilized (myr+)MAPP oligomers through disulfide bridges. In the absence of liposomes, we observed only the monomeric fraction, but in their presence, we also observed dimers, trimers, and even tetramers (*Figure 2B*). Our previous research indicated that (myr−)MA at the same concentration primarily forms dimers and some trimers (*Prchal et al., 2012*). In the T69C trimer, C69 is located directly opposite C62 from the second monomer (*Figure 2A*). Dimers are likely stabilized by disulfide bridges between C62 residues and in the T69C mutant also by two disulfide bridges between C42 and C69 (*Figure 5*). We also observed tetramers, which we believe are two non-specifically cross-linked dimers.

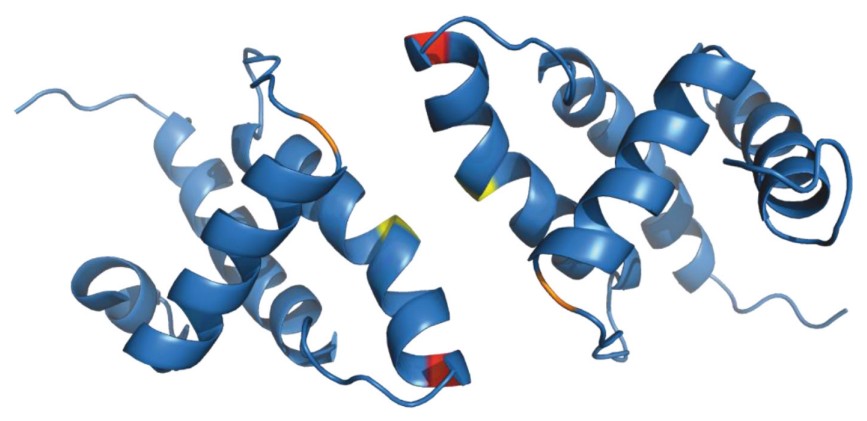

**Figure 5.** Structure of (myr−)MA wild-type dimer (*Vlach et al., 2009*). Previously published ribbon structure of (myr−)MA wild-type dimer (*Vlach et al., 2009*) showing positions of residues T69 (in red), C62 (in yelow), and C42 (in orange).

Our findings reveal that the PM interaction induces myristoyl exposure from the hydrophobic core of M-PMV MA. The myristoyl switch then supports both proteolytic separation of MA from the downstream part of Gag and MA oligomerization. Broadly, these results suggest that similar mechanisms of functional modulation may occur in domains of other myristoylated proteins. For example, recoverin and calcium- and integrin-binding protein 2 exploit a myristoyl switch to modulate calcium binding (*Ames et al., 1997*). Thus, myristoyl exposure-triggered structural changes may have more general validity and may play various regulatory roles in other N-terminally myristoylated proteins.

## Materials and methods

### Vectors

All mutations in the vector pET22bMAPPHis (*Prchal et al., 2011*) were introduced using Efficient Mutagenesis Independent of Ligation (EMILI) (*Füzik et al., 2014*). The mutation responsible for amino acid substitution I51A was introduced using primers MAPPHisI51Af GGAACCGCAGATATTAAACG GTGGCGTAGAG and MAPPHisI51Ar CGTTTAATATCTGCGGTTCCCTCTTGCGG, for the amino acid substitution A79V using primers MAPPHisA79Vf TAACTGTTTTCTCTTACTGGAACTTAATTAAAGAAT TGATAGATAAG and MAPPHisA79Vr TAAGAGAAAACAGTTACTGGGACTTTCTCCG, and for amino acid substitution T69C using primers MAPPHisT69Cf TATTACAATTGTTTTGGCCCGGAGAAAGTCCC and MAPPHisT69Cr GGGCCAAAACAATTGTAATAGTCTTGGAAACAGTCG.

### Recombinant protein production and purification

All (myr−) and (myr+)MAPPs were produced in *E. coli* BL21 (DE3) cultivated in LB medium and purified using metal affinity chromatography on Ni-NTA agarose according to a previously published protocol (*Prchal et al., 2011*). The (myr+)MAPP T69C protein was purified under reducing conditions (buffers containing 0,05% mercaptoethanol). Uniformly isotopically labeled proteins were produced using M9 minimal medium (*Sambrook et al., 2001*) with [U-$^{5}$N]NH$_4$Cl and D-[U-$^{13}$C]glucose (CIL, USA). The identities of the prepared proteins and the degrees of myristoylation were confirmed using MALDI (Matrix Assisted Laser Desorption/Ionization-Time Of Flight)-TOF/TOF mass spectrometry on an Autoflex speed mass spectrometer (Bruker Daltonics). The 13 kDa form of M-PMV protease was prepared using a previously published protocol (*Zábranský et al., 1998*).

### Liposome preparation

To prepare liposomes mimicking the PM inner leaflet (*Doktorova et al., 2017*), individual lipids (purchased from Avanti Polar Lipids, Inc) were dissolved in chloroform or in chloroform/methanol/water (20:9:1) in the case of PI(4,5)P$_2$ and thoroughly mixed to a final lipid concentration of 5 mg/ml. To obtain 250 µl of lipid mixture, we mixed 310 µg of cholesterol, 400 µg of phosphatidylethanolamine, 75 µg of phosphatidylcholine, 290 µg of phosphatidylserine, 38 µg of PI(4,5)P$_2$, and 140 µg of phosphatidylinositol. Chloroform was evaporated, and the lipid mixture was resuspended in a protease cleavage buffer or in phosphate-buffered saline (PBS). Liposomes were formed using a mini-extruder (Avanti Polar Lipids, Inc) with a 100 nm polycarbonate filter.

### MAPP cleavage by M-PMV protease

All (myr−) and (myr+)MAPP proteins were cleaved by M-PMV protease in the protease cleavage buffer (50 mM acetate, pH 5.3, 300 mM NaCl, 0.05% mercaptoethanol). Briefly, 80 µg of each protein solution was incubated with 2 U of protease (1 U of M-PMV protease cleaves 100 µg of (myr−)MAPP in 1 hr) in a total volume of 220 µl of protease cleavage buffer in the absence or presence of 20 µl liposomes. Aliquots (20 µl each) were collected at time intervals of 1, 2, 4, and 24 hr, resuspended in 2× reducing protein loading buffer (PLB), and analyzed by Tris-Tricine sodium dodecyl sulfate–polyacrylamide gel electrophoresis (SDS–PAGE).

### Interaction of (myr+)MAPP T69C with liposomes

A 40 µg portion of (myr+)MAPP T69C solution in PBS (137 mM NaCl, 2.7 mM KCl, 10 mM Na$_2$HPO$_4$·2H$_2$O, 2 mM KH$_2$PO$_4$, pH 7.4, reducing PBS contains 0.05% mercaptoethanol) was mixed with liposome suspension in a 1:1 (vol/vol) ratio to obtain a total volume of 100 µl. Protein diluted 1:1 with reducing PBS was used as a control. The samples were dialyzed overnight either against

non-reducing or reducing PBS. Aliquots were collected, resuspended in 2× non-reducing PLB, and analyzed by Tris-Tricine SDS–PAGE.

## Hydrogen–deuterium exchange

For HDX labeling experiments, 0.2 mM wild-type (WT) (myr−)MAPP, WT (myr+)MAPP, and the I51A, I86A, A79L, and A79V mutants were mixed with $D_2O$ in a 1:9 ratio. Samples were incubated at 4°C for 0, 2, 5, 10, 20, or 120 s and quenched with an equal volume of quench buffer (8 M urea, 1 M glycine, pH 2.51). In case of the (myr+)MAPP I51A mutant, 200 mM TCEP (Tris(2-carboxyethyl)phosphine) was added to the quench buffer to increase its sequence coverage. The HDX experiments were performed in triplicates for each labeling time point.

## Liquid chromatography–mass spectrometry analysis

Peptides were identified by tandem MS of non-deuterated proteins. Samples were injected into a refrigerated UPLC (Ultra Performance Liquid Chromatography) system (NanoAcquity, Waters) with chromatographic elements held at 0°C. The samples were then passed through a pepsin protease column (Enzymate Protein Pepsin Column, 300 Å, 5 µm, 2.1 mm × 30 mm, Waters) at 15°C and flow rate of 100 µl min$^{-1}$ (0.1% vol/vol FA), and the generated peptides were trapped and desalted for 3 min on a VanGuard pre-column (2.1 mm × 5 mm Waters Acquity UPLC BEh $C_{18}$ (pore size 1.7 µm)). Peptides were then separated in gradient of acetonitrile for 12 min over an Acquity UPLC column (1 mm × 100 mm, 1.7 µm BEH $C_{18}$) for 12 min (10–35% $CH_3CN$ vol/vol and 0.1% vol/vol FA, flow rate 40 µl min$^{-1}$). The MS spectra were acquired with a Synapt G2 mass spectrometer (ESI-Q/TOF; Waters) in mass range from 50 to 2000 $m/z$ with Leu-enkephalin serving as a continuous (lock-spray) calibration standard and performing a scan every 0.4 s. The list of peptides was obtained by using ProteinLynx Global SERVER (PLGS; Waters) version 3.0.2 with processing parameters as follows: chromatographic peak width – automatic, MS TOF (Mass Spectrometry Time Of Flight) resolution – automatic, lock mass for charge +1 – 556.2771 Da/e, lock mass window – 0.25 Da, low energy threshold – 135.0 counts, elevated energy threshold – 30.0 counts, and intensity threshold – 750.0 counts. PLGS work-flow parameters were as follows: searching against fasta file containing forward and reverse sequences of examined proteins and Pepsin (Uniprot code P00791), peptide tolerance and fragment tolerance – automatic, minimum fragment ion matches per peptide – 3, minimum fragment ion matches per protein – 7, minimum peptide matches per protein – 1, primary digest reagent – non-specific, number of missed cleavages 3, oxidation of methionines as a variable modifier reagent, false discovery rate 5, monoisotopic mass of peptides with charge +1. The LC analysis of labeled samples was identical to that of non-deuterated samples.

DynamX 3.0 (Waters) was used to filter peptides for the determination of HDX differences between examined proteins by selecting those having the length up to 25 amino acids, presenting 0.3 fragment per amino acid and showing mass error for the precursor ion below 10 ppm. In addition, only the peptides that were identified in at least three out of five of the acquired MS/MS files and having a minimum signal intensity 3000 with retention time RSD (Relative Standard Deviation) up to 5% were used for further analysis. MS Files were processed according to the parameters as follow: both chromatographic peak width and MS TOF resolution as automatic, 556.2771 Da as a lock mass for charge +1, lock mass window 0.25 Da, low energy threshold 130, and elution time range to search in for the data 2.5–9 min. DynamX advanced processing parameters were not applied.

## NMR spectroscopy

All NMR data were collected on a Bruker AvanceIII 600-MHz NMR spectrometer equipped with a cryoprobe (Bruker BioSpin, GmbH, Germany). The backbone atoms of myrMA were assigned using the standard set of triple-resonance experiments (HNCA, HN(CO)CA, HNCACB, CBCA(CO)NH, and HNCO). An estimate of CSI and the backbone dihedral angles $\Phi$ and $\Psi$ was performed with TALOS+ software and was based on the $^1HN$, $^{13}CO$, $^{13}C\alpha$, $^{13}C\beta$, and $^{15}NH$ chemical shifts. The data were processed with TopSpin (Bruker BioSpin GmbH, version 3.6) and analyzed using CcpNmr analysis (*Vranken et al., 2005*). Structures were visualized with the PyMOL Molecular Graphics System (Version 2.0, Schrödinger, LLC).

## Acknowledgements

The research was supported by the Czech Science Foundation (grant No. 22-19250S) and by the project National Institute of Virology and Bacteriology (Programme EXCELES, ID Project No. LX22NPO5103) – Funded by the European Union – Next Generation EU. We would like to thank Petra Junkova for careful reading and suggestions to improve the manuscript.

## Additional information

### Funding

| Funder | Grant reference number | Author |
|---|---|---|
| Grant agency of the Czech Republic | 22-19250S | Tomas Ruml |
| European Union - Next Generation EU | Programme EXCELES, ID Project No. LX22NPO5103 | Tomas Ruml |

The funders had no role in study design, data collection, and interpretation, or the decision to submit the work for publication.

### Author contributions
Markéta Častorálová, Jakub Sýs, Jan Prchal, Data curation, Formal analysis, Investigation, Visualization, Methodology, Writing - original draft; Anna Pavlů, Lucie Prokopová, Zina Briki, Investigation; Martin Hubálek, Formal analysis, Supervision; Tomas Ruml, Conceptualization, Data curation, Formal analysis, Supervision, Funding acquisition, Validation, Investigation, Visualization, Methodology, Writing - original draft, Project administration, Writing - review and editing

### Author ORCIDs
Jakub Sýs ⓘ http://orcid.org/0000-0003-2589-1631
Jan Prchal ⓘ http://orcid.org/0000-0002-3398-5059
Martin Hubálek ⓘ http://orcid.org/0000-0003-0247-7956
Tomas Ruml ⓘ http://orcid.org/0000-0002-5698-4366

### Decision letter and Author response
Decision letter https://doi.org/10.7554/eLife.93489.sa1
Author response https://doi.org/10.7554/eLife.93489.sa2

## Additional files

### Supplementary files
• MDAR checklist

• Supplementary file 1. Comparison of observed and calculated backbone atoms chemical shifts for (myr+)MAPP and (myr-)MAPP.

• Supplementary file 2. Estimated backbone order parameters and secondary structure for (myr+) MAPP and (myr-)MAPP.

### Data availability
The data were deposited in Dryad under the DOI: https://doi.org/10.5061/dryad.c59zw3rfn.

The following dataset was generated:

| Author(s) | Year | Dataset title | Dataset URL | Database and Identifier |
|---|---|---|---|---|
| Castoralova M, Sys J, Prchal J, Pavlu A, Prokopova L, Briki Z, Hubalek M, Ruml T | 2023 | Data from: A myristoyl switch at the plasma membrane triggers cleavage and oligomerization of Mason-Pfizer monkey virus matrix protein | https://doi.org/10.5061/dryad.c59zw3rfn | Dryad Digital Repository, 10.5061/dryad.c59zw3rfn |

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
