## [Editor Report]

This valuable study advances our understanding of how the viral protease in a D-type retrovirus is activated and in particular how the exposure of the myristoyl group is required for processing of the Gag matrix precursor. The supporting evidence is convincing and suggests that M-PMV proteolytic maturation is triggered by a myristyl switch that occurs when the matrix (MA) domain of Gag interacts with the PM. This manuscript is of interest to retrovirologists and structural biologists.

---

## [Decision Letter]

**Decision letter after peer review:**

Thank you for submitting your article "A myristoyl switch at the plasma membrane triggers cleavage and oligomerization of Mason-Pfizer monkey virus matrix protein" for consideration by *eLife*. Your article has been reviewed by 3 peer reviewers, and the evaluation has been overseen by a Reviewing Editor and Miles Davenport as the Senior Editor. The reviewers have opted to remain anonymous.

Essential revisions:

1) All recommendations by reviewers #1 and #2.

2) With regards to reviewer #3, the most important revision is: "The authors can show the specificity of the oligomerization reaction by moving the cysteine to other surface locations and testing whether other locations would also support oligomer formation. Since the diffusion of the MA on the liposome is 2D, it is possible that oligomers would form with almost any surface interactions, which would be informative. Regarding the authors extrapolation of the observed allosteric interaction within monomers in solution, to the fully assembled particles interacting with membrane, I think all they have to do is discuss it. Comparison to recent MA maturation in HIV-1 observed by CryoEM would be nice to add. "

*Reviewer #1 (Recommendations for the authors):*

1. Figure 1. For any data of this type, the reader will want to know how reproducible the results are, i.e., how many times the experiment was repeated and whether quantification from the individual repeats demonstrates that the differences observed are statistically significant.

2. Figure 3. Many general readers will not be familiar with HDX. The data in this figure should be explained in more detail for such readers.

3. Similarly, in Figure 4, no explanation is provided about what is being shown and what the Talos+ software does. More explanation should be provided.

*Reviewer #2 (Recommendations for the authors):*

I have only the most trivial criticisms of the manuscript. Line 317 contains an error producing "the was". I don't believe "reducing PBS" (line 451) is ever defined.

*Reviewer #3 (Recommendations for the authors):*

1. The manuscript is very interesting but fails to provide evidence for the claims in the title. Consider reworking the title so it is supported by the presented data.

2. The evidence of oligomerization is very limited with no controls aside from the non-reducing conditions. Please provide more direct evidence of this claim by making more controls.

3. There is no evidence presented that shows the proposed conformational changes in the matrix actually happen during interaction of the fully assembled particle by the plasma membrane. Specifically the conformational changes are observed by deuterium exchange experiments on mutants which we presume affect myristol exposure in the absence of membrane which is rather weak.

4. Regarding the authors extrapolation of the observed allosteric interaction within monomers in solution, to the fully assembled particles interacting with membrane, I think all they have to do is discuss it. Comparison to recent MA maturation in HIV-1 observed by CryoEM would be nice to add.

---

## [Author Response]

Essential revisions:1) All recommendations by reviewers #1 and #2.2) With regards to reviewer #3, the most important revision is: "The authors can show the specificity of the oligomerization reaction by moving the cysteine to other surface locations and testing whether other locations would also support oligomer formation. Since the diffusion of the MA on the liposome is 2D, it is possible that oligomers would form with almost any surface interactions, which would be informative.

We performed another experiment comparing the oligomerization of WT (myr+)MAPP containing two cysteines presumably capable of disulfide bridge formation with the (myr+)MAPP T69C mutant. The resulting gel shows that the mere presence of cysteines at positions other than the trimerization interface is not sufficient to promote oligomerization upon interaction with liposomes.

Below is the text added to the Results section:

“Notably, the WT (myr+)MAPP, which features two surface-exposed cysteines (C42 and C62), does not undergo oligomerization under identical conditions (Figure 2—figure supplement 1). This suggests, that random cysteine exposure does not lead to the formation of disulfide bridges in the WT (myr+)MAPP, indicating that the presence of surface-exposed cysteines is insufficient to induce non-specific oligomerization. This is in contrast to the (myr+)MAPP T69C, where oligomerization is supported by the addition of cysteine in position 69 (Figure 2, A – red) which can form disulfide bridge with cysteine 62 (Figure 2, A – yellow).”

Regarding the authors extrapolation of the observed allosteric interaction within monomers in solution, to the fully assembled particles interacting with membrane, I think all they have to do is discuss it. Comparison to recent MA maturation in HIV-1 observed by CryoEM would be nice to add. "

We have added the following paragraph to the manuscript Discussion in order to justify the appropriateness of our approach:

M-PMV significantly differs from the C-type retroviruses in the MA interaction with membrane components, Gag organization and particle assembly. Some of these properties preclude the use of full-length M-PMV Gag for structural and some functional studies. The obstacles are related to the fact that M-PMV Gag assembles into immature particles in vitro in the absence of membranes (Klikova et al., 1995), but HDX and NMR methods are not suitable for working with pre-assembled particles. A possible method for obtaining relevant structural data would be cryo-EM, which was used to obtain the high-resolution lattice structure of MA HIV-1 (Qu et al., 2021). However, unlike the distinctive hexagonal lattice formed by HIV-1 MA trimers, the M-PMV MA layer lacks any observable ordered structure (James Stacey, personal communication). This confirms lower oligomerization capacity of M-PMV MA compared to HIV-1 MA. Given the above, the MA-PP fusion protein appears to be the best available model mimicking M-PMV. We also hypothesize that within the overall context of Gag, M-PMV MA retains significantly higher structural flexibility compared to HIV-1 MA. This is because HIV-1 MA is separated from the rigid CA Gag domain layer solely by relatively short flexible linker, in contrast to M-PMV MA which is separated from CA domain by two protein domains, PP and p12.

Reviewer #1 (Recommendations for the authors):1. Figure 1. For any data of this type, the reader will want to know how reproducible the results are, i.e., how many times the experiment was repeated and whether quantification from the individual repeats demonstrates that the differences observed are statistically significant.

We added the following sentence to the figure lagend:

“All the experiments were performed in three biological replicates with the same results.”

2. Figure 3. Many general readers will not be familiar with HDX. The data in this figure should be explained in more detail for such readers.

We thank the reviewer for the suggestion. The following paragraph was added to the manuscript:

“HDX is a valuable method for probing protein conformation and dynamics without perturbation. This technique utilizes mass spectrometry (MS) to monitor the time-dependent incorporation of deuterium into solvent-accessible backbone amide hydrogens of peptides. Rapid exchange rates of these amide hydrogens suggest flexibility in the surrounding region, whereas slower exchange rates indicate a relatively rigid conformation. Consequently, these exchange rates provide insights into the protein conformation.”

3. Similarly, in Figure 4, no explanation is provided about what is being shown and what the Talos+ software does. More explanation should be provided.

We added following explanation to the manuscript:

“The chemical shift of the backbone atom of a protein strongly depends on the local structure.

We then used TALOS+ software (Shen et al., 2009), which compares the observed chemical shifts of backbone atoms with a large database of proteins with known structures to predict the secondary structure of this region.”

Reviewer #2 (Recommendations for the authors):I have only the most trivial criticisms of the manuscript. Line 317 contains an error producing "the was". I don't believe "reducing PBS" (line 451) is ever defined.

We have clarified the reducing PBS by modifying the sentence in the Materials and methods section as follows:

“A 40 µg portion of (myr+)MAPP T69C solution in PBS (137 mM NaCl, 2.7 mM KCl, 10 mM Na_2_HPO_4_·2H_2_O, 2 mM KH_2_PO_4_, pH 7.4, reducing PBS contains 0,05% mercaptoethanol)…”

Reviewer #3 (Recommendations for the authors):1. The manuscript is very interesting but fails to provide evidence for the claims in the title. Consider reworking the title so it is supported by the presented data.2. The evidence of oligomerization is very limited with no controls aside from the non-reducing conditions. Please provide more direct evidence of this claim by making more controls.

We performed another experiment comparing the oligomerization of WT (myr+)MAPP containing two cysteines presumably capable of disulfide bridge formation with the (myr+)MAPP T69C mutant. The resulting gel shows that the mere presence of cysteines at positions other than the trimerization interface is not sufficient to promote oligomerization upon interaction with liposomes.

Below is the text added to the Results section:

“Notably, the WT (myr+)MAPP, which features two surface-exposed cysteines, does not undergo oligomerization under identical conditions (Figure 2—figure supplement 1). This suggests that random cysteine exposure does not lead to the formation of disulfide bridges in the WT (myr+)MAPP, indication that it is insufficient to induce non-specific oligomerization. This is in contrast to the (myr+)MAPP T69C, where oligomerization of is supported by the presence of two cysteines situated proximally at the trimerization as shown in Figure 2.”

3. There is no evidence presented that shows the proposed conformational changes in the matrix actually happen during interaction of the fully assembled particle by the plasma membrane. Specifically the conformational changes are observed by deuterium exchange experiments on mutants which we presume affect myristol exposure in the absence of membrane which is rather weak.4. Regarding the authors extrapolation of the observed allosteric interaction within monomers in solution, to the fully assembled particles interacting with membrane, I think all they have to do is discuss it. Comparison to recent MA maturation in HIV-1 observed by CryoEM would be nice to add.

Combined response to both comments:

We agree with the comments that the experiments were not performed with the fully assembled particles interacting with the plasma membrane. We have added the following paragraph to the discussion, reflecting comments 3 and 4, to justify the appropriateness of our approach.

M-PMV significantly differs from the C-type retroviruses in the MA interaction with membrane components, Gag organization and particle assembly. Some of these properties preclude the use of full-length M-PMV Gag for structural and some functional studies. The obstacles are related to the fact that M-PMV Gag assembles into immature particles in vitro in the absence of membranes (Klikova et al., 1995), but HDX and NMR methods are not suitable for working with pre-assembled particles. A possible method for obtaining relevant structural data would be cryo-EM, which was used to obtain the high-resolution lattice structure of MA HIV-1 (Qu et al., 2021). However, unlike the distinctive hexagonal lattice formed by HIV-1 MA trimers, the M-PMV MA layer lacks any observable ordered structure (James Stacey, personal communication). This confirms lower oligomerization capacity of M-PMV MA compared to HIV-1 MA. Given the above, the MA-PP fusion protein appears to be the best available model mimicking M-PMV. We also hypothesize that within the overall context of Gag, M-PMV MA retains significantly higher structural flexibility compared to HIV-1 MA. This is because HIV-1 MA is separated from the rigid CA Gag domain layer solely by relatively short flexible linker, in contrast to M-PMV MA which is separated from CA domain by two protein domains, PP and p12.